# Exploration of the Cs Trapping Phenomenon by Combining Graphene Oxide with α-K_6_P_2_W_18_O_62_ as Nanocomposite

**DOI:** 10.3390/ma14195577

**Published:** 2021-09-26

**Authors:** Bangun Satrio Nugroho, Akane Kato, Chie Kowa, Tomoya Nakashima, Atsushi Wada, Muh. Nur Khoiru Wihadi, Satoru Nakashima

**Affiliations:** 1Radioactivity Environmental Protection Course, Phoenix Leader Education Program, Hiroshima University, 1-1-1 Kagamiyama, Higashi-Hiroshima 739-8524, Japan; d192886@hiroshima-u.ac.jp; 2Department of Chemistry, Graduate School of Science, Hiroshima University, 1-3-1 Kagamiyama, Higashi-Hiroshima 739-8526, Japan; kato.akane@jaea.go.jp (A.K.); c.k_0715_titty@icloud.com (C.K.); hakatomo0601@gmail.com (T.N.); m204479@hiroshima-u.ac.jp (A.W.); 3Department of Chemistry, Faculty of Military Mathematics and Natural Science, The Republic of Indonesia Defense University, Bogor 16810, Indonesia; nur.wihadi@idu.ac.id; 4National Research and Innovation Agency (Badan Riset dan Inovasi Nasional) of The Republic of Indonesia, Kawasan Puspiptek, Serpong, Tangerang 15314, Indonesia; 5Natural Science Center for Basic Research and Development, Hiroshima University, 1-4-2 Kagamiyama, Higashi-Hiroshima 739-8526, Japan

**Keywords:** graphene oxide, Dawson-type polyoxometalate, cesium, sp^2^/sp^3^ carbon domain, nanocomposite

## Abstract

A graphene oxide-based α-K_6_P_2_W_18_O_62_ (Dawson-type polyoxometalate) nanocomposite was formed by using two types of graphene oxide (GO) samples with different C/O compositions. Herein, based on the interaction of GO, polyoxometalates (POMs), and their nanocomposites with the Cs cation, quantitative data have been provided to explicate the morphology and Cs adsorption character. The morphology of the GO-POM nanocomposites was characterized by using TEM and SEM imaging. These results show that the POM particle successfully interacted above the surface of GO. The imaging also captured many small black spots on the surface of the nanocomposite after Cs adsorption. Furthermore, ICP-AES, the PXRD pattern, IR spectra, and Raman spectra all emphasized that the Cs adsorption occurred. The adsorption occurred by an aggregation process. Furthermore, the difference in the C/O ratio in each GO sample indicated that the ratio has significantly influenced the character of the GO-POM nanocomposite for the Cs adsorption. It was shown that the oxidized zone (sp^2^/sp^3^ hybrid carbon) of each nanocomposite sample was enlarged by forming the nanocomposite compared to the corresponding original GO sample. The Cs adsorption performance was also influenced after forming a composite. The present study also exhibited the fact that the sharp and intense diffractions in the PXRD were significantly reduced after the Cs adsorption. The result highlights that the interlayer distance was changed after Cs adsorption in all nanocomposite samples. This has a good correlation with the Raman spectra in which the second-order peaks changed after Cs adsorption.

## 1. Introduction

The destruction of the reactor cores of the Fukushima Daiichi Nuclear Power Plant (FDNPP) ten years ago has had a large influence on the environment. Many researchers predicted the quantity of released material [1] and modeling of the radiation fallout immediately [2]. It is important to consider the radioactive isotopes delivered to the agriculture area [3], tap water [4], and forest area [5]. The subject of considerable concern for the people is the possibility of the radiation exposure to their body. Both the external exposure and internal exposure have radiological consequences that can cause cancer [6,7]. Therefore, it is an essential task to treat and decontaminate the contaminated environment.

^137^Cs, which has a half-life of 30 years, is one of the nuclides that has been found in the contaminated area [8]. The ^137^Cs originated from the Tokyo Electric Power Company (TEPCO) FDNPP accident [9]. Many studies have investigated how to handle radioactive cesium as a crucial radioisotope [10]. Chemical adsorption methods that use composites [11] and clay [12] to catch the Cs are the techniques suggested in many papers. According to these methods, the adsorption mechanism emerges as an important concept (such as the hydroxyl-interlayered vermiculate, 3D microporous composite) and encourages the synthesis of new material. Therefore, the goal of this work is to create a new sorbent material, which has an ability to adsorb the Cs with high efficiency. As is well known, the skeleton of GO having different oxygen functional groups can be considered as a bonding point [13]. Based on that, the chemical modification opens up the idea for tailoring the graphene oxide (GO) with another compound.

Zilong Liu et al. (2019) investigated GO as a novel carbonaceous material and found that the GO surface has an interesting interaction using a polar -COO(H) and non-polar (-CH_3_) group in order to adsorb different kinds of cations (charge, size, complexing capability) by a proposed ion-bridging mechanism [14]. Another important thing is the solubility of GO in the diverse range of solvents, including water due to the hydroxyl, epoxide functional group and carboxyl group attached on the surface and the sheet edge [15,16].

On the other hand, polyoxometalates (POMs) provide an excellent, robust, and discrete material, which has reversible multi-electron redox properties [17]. Some researchers have employed POMs as a potential Cs adsorption material by using a cation exchange mechanism [18]. For example, it was investigated that the interaction of Cs^+^ with POMs makes it easier to enter and diffuse through the solid-state structure compared to other types of cations by considering the dehydration enthalpy and hydration radius of the cations [19]. The K_6_P_2_W_18_O_62_ as a Dawson-type compound has an anionic molecular framework constructed by two phosphates, 18 tungstates surrounding a central potassium. POM has a large molecular size [20] and high potential to reduce [17], and it is a structurally well-defined oxide cluster anion. Furthermore, R.D. Gall et al. confirmed that POM particles could easily be adsorbed on the carbon material [21]. It is also proved by previous study that the anionic PW_12_ cluster can be adsorbed on the reduced GO and can improve the dispersibility of water on the reduced graphene sheet [22].

Considering the tremendous properties of GO and α-K_6_P_2_W_18_O_62_, it is highly desirable to synthesize a new nanocomposite with Cs adsorption ability using the GO and POMs [23]. In this study, two types of GO samples were employed. One is GO, which has a large amount of carbon (C), and the other is GO, which has a large amount of oxygen (O). They have a variant sp^2^/sp^3^ structure. These GOs can be produced by controlling the oxidation degree, which depends on several factors such as the reaction condition, oxidizing agent, and graphite source [24]. It has been known that the GO structure is not precisely determined because of a variety of local arrangements of functional groups [25,26]. Each GO sample was incorporated with α-K_6_P_2_W_18_O_62_. We investigated the interaction of Cs^+^ with the nanocomposite by characterizing the surface structure of the composite material and also by considering the adsorption capacity of the nanocomposite for radioactive Cs. Each property in each material is expected to synergistically strengthen the adsorption capacity of the nanocomposite. A high Cs adsorption capacity was clearly achieved by using the nanocomposite.

## 2. Materials and Methods

### 2.1. Materials

All raw chemicals were reagent grade and were used without purification, and distilled water was used in all experiments. Graphite, Na_2_WO_4_·2H_2_O, CsCl, and methanol were purchased from Sigma Aldrich Chemistry. NaNO_3_, H_2_SO_4_, HCl, H_3_PO_4_, and KCl were purchased from Fuji Film Wako Pure Chemical Corporation. KMnO_4_ and H_2_O_2_ were purchased from the Tokyo Chemical Industry.

### 2.2. Characterizations

Fourier transform infrared spectroscopy (FT-720, HORIBA, Ltd., Kyoto, Japan), powder X-ray diffraction measurement (Rigaku, Thermo plus, XRD-DSC II, Tokyo, Japan), and Raman spectroscopy (Raman spectrometer HORIBA, Ltd., T64000, Kyoto, Japan) were conducted to characterize the functional groups, crystallinity, number of layers of GO, size of GO samples, interlayer distance, and the quality of samples. Scanning electron microscope (SEM) (ultra-high resolution field emission scanning electron microscope device (Hitachi High-Technologies Corporation, S-5200, Tokyo, Japan)), EDAX (Genesis XM2, Tokyo, Japan), transmission electron microscopy (TEM) (ultra-high resolution transmission electron microscope (JEOL Ltd., JEM-2010, Tokyo, Japan)) were performed to collect imaging data of the samples. ICP-AES (atomic emission spectroscopy) (SPS3500, SII Nanotechnology Inc., Tokyo, Japan) was used to calculate the adsorption capability of the samples.

### 2.3. Synthesis of Graphene Oxide

All GO samples were synthesized using the modified Hummers method [27]. A pre-cooling procedure was implemented by keeping all the reagents and solvent in the refrigerator for one night before the synthesis process [28]. In order to start the synthesis process, first, a graphite (1.0 g) and NaNO_3_ (1.5 g) were put into the vial. Then, a concentrated H_2_SO_4_ (50 mL) was added carefully. We stirred the mixture for 30 min at room temperature. In this step, the color was blackish green. Then, we put the mixture in the refrigerator for 30 min to keep it in the cooling condition below 10 °C. After that, going to the critical step, we added KMnO_4_ 4.0 g slowly and carefully. Then, we stirred it for about 20 min and then set the temperature at 35 °C under stirring for 2 h. After this reaction, the color of the solution became brown. We diluted the mixture with 80 mL of distilled water with stirring for 15 min. Then, we continued stirring for the other 30 min at room temperature. We added 150 mL of distilled water slowly and carefully with stirring for 15 min. After that, 12 mL of 30% H_2_O_2_ was dropped carefully to the solution. A large number of bubbles were released. The solution became golden yellow. The solid particle was filtered, washed using 5% HCl, and followed by using distilled water. We dried the solid at 60 °C for 24 h. In order to distinguish each GO sample, we termed these GO samples by index number, which have different amounts of C based on elemental analysis. There are two types of GO sample based on C/O composition. The GO_c70_ and GO_c72_ have 70–72 wt % C element, while GO_c39_ and GO_c40_ have ≈ 39 wt % C element.

### 2.4. Synthesis of α-K_6_P_2_W_18_O_62_

The α-K_6_P_2_W_18_O_62_ was synthesized according to the modified Nadjo method [29]. The synthesis route started with dissolving the starting material, Na_2_WO_4_·2H_2_O 15 g into 17.5 mL of distilled water. We vigorously stirred for 15 min until the cloudy solution became limpid. This was followed by an acidification process: HCl 4M (12.5 mL) was added dropwise for 45 min carefully. In this process, white precipitate was formed. In order to avoid agglomeration and to disperse homogenously during the acidification process, the vigorous stirring was performed. We checked the pH to be around 6–7. Then, 2M H_3_PO_4_ (12.5 mL) was added, and we checked the pH to become around 2–3. The solution was refluxed for 24 h; then this solution was cooled at room temperature for a while. Then, 7.5 g of KCl was added to the solution. Afterwards, this solution was filtered off by paper filtration, and the crude material was obtained. Then, we dissolved it into 32.5 mL of distilled water, and the solution became limpid. The next reflux process was performed at 80 °C for 72 h. Then, it was cooled at room temperature. The last step was keeping the solution in the refrigerator, and after 24 h, the well-behaved crystals of α-K_6_P_2_W_18_O_62_ were produced (6.5 g).

### 2.5. Synthesis of GO-POM Composite

The composite sample was distinguished into eight types of composite. Each composite was produced by incorporating the GO sample (GO_c70_, GO_c72_, GO_c39_, and GO_c40_) with Dawson-type POM. The GO-POM composite was generated in two different concentration ratios (concentration ratios of GO:POM: 1:8 and 4:1). The constitution is summarized in Section 3.7.2. These treatments (different concentration ratios of GO:POM) were employed in order to investigate the role of GO and Dawson-type POM in the Cs adsorption and to know the proper composition. In this synthesis process, mixed solvent was used with the ratio 1:1 (distilled water/methanol). The GO aqueous dispersion in the first concentration condition (1:8) was prepared by dispersing 1 g of GO into the solvent (100 mL) with sonification treatment for 5 min. Then, we prepared the POM solution by dissolving 1 g of α-K_6_P_2_W_18_O_62_ into the solvent 12.5 mL. Afterwards, we poured the POM solution into the GO suspension vial, added diluted HCl to adjust the pH in the range 2–3, and then vigorous stirred for 24 h. In the second concentration condition (4:1), we prepared the GO solution by dispersing GO (0.5 g) into the solvent (25 mL) with a sonification treatment for 5 min. The POM solution was prepared by dissolving the α-K_6_P_2_W_18_O_62_ 0.5 g into the solvent (100 mL). The POM solution was poured into the GO suspension; then, we added diluted HCl to adjust the pH in the range 2–3 and after that stirred for 24 h. Finally, the solid material was collected by using filter paper and dried at 90 °C for 24 h. Then, the composite product was indexed based on the value of the C element of GO (c70, c72, c39, and c40 due to elemental analysis) and the concentration ratio of GO/POM (1:8 and 4:1) in order to distinguish each composite’s name, as follows: [GO_70_POM]_18_, [GO_70_POM]_41_, [GO_72_POM]_18_, [GO_72_POM]_41_, [GO_39_POM]_18_, [GO_39_POM]_41_, [GO_40_POM]_18_, and [GO_40_POM]_41_. Contant’s group reported the hydrolytic stability of Dawson-type POM depending on the acidity of the solution. That is, Dawson-type POM was stable at pH lower than 6, and when the pH increased above 6, the formation of lacunary species occurs [30]. In the case of our system, preparation and adsorption experiments were performed at pH lower than 5. Therefore, the Dawson-type POM was stable in the present solution.

### 2.6. Adsorption Experiment

The cesium adsorption was conducted as below. First, 0.08 g of each sample (POM, GO samples, GO-POM composites) was dispersed into 30 mL of distilled water with sonification treatment for 5 min. Second, 0.6 g of CsCl (3.6 mmol) was added to the solution. Then, the mixture was stirred for 24 h.

The performance of Cs adsorption was analyzed by using ICP optical emission spectroscopy. The solid adsorbent was separated from the solution by using paper filtration. The measurement was conducted in duplicate in each sample solution. The adsorption efficiency was calculated by Equation (1), and the adsorption capacity was calculated by Equation (2).
(1)$$\begin{array}{l} Adsorption\;efficiency\;\left(\%\;Ads.eff.\right)=\frac{\left(C_{0}-C_{t}\right)}{C_{0}}\times 100 \end{array}$$
(2)$$\begin{array}{l} Adsorption\;capacity=\frac{\%Ads.eff.\times C_{0}}{W} \end{array}$$
where *C*_0_ (mmol/L) is the initial concentration of Cs, and *C_t_* (mmol/L) is the remained concentration of Cs in the solution after treatment. *W* (g) is the dry weight of the adsorbent used in the adsorption experiment.

## 3. Results and Discussion

### 3.1. Elemental Analysis

Table 1 shows the results of elemental analysis of GO_c70_, GO_c72_, GO_c39_, and GO_c40_. It can be seen that the GO_c70_ and GO_c72_ have 70–72 wt % C element, while GO_c39_ and GO_c40_ have ≈39 wt % C element. In the synthesis process, there is one thing that might make the sample a bit different. In the treatment of KMnO_4_, especially for GO_c39_ and GO_c40_, the KMnO_4_ was treated more carefully (one minute stop, one minute go). It was shown that the degree of oxidation was controlled by the controlled oxidation process. In particular, it can be assumed that there is a difference in structure between the two types of GO. GO_c70_ and GO_c72_ indicating few oxygen defects in the structure have an important contrast with GO_c39_ and GO_c40._ However, deeper characterization is still required, since the oxygen defect is not only the case. As emphasized in the previous literature, there is no evident relation between the oxidation degree and ordered domain that is exhibited in the Raman spectrum [31].

### 3.2. FT-IR Spectra

Appendix A show the FT-IR spectra of GO and Dawson-type compounds. The Dawson-type POM was successfully produced by the Nadjo synthesis route. The IR absorptions of the present α-K_6_P_2_W_18_O_62_ are similar with those of the literature [29]. All the GO samples exhibit a broad absorption band of O-H stretching vibration around 3400 cm^−1^, C=C bond stretching at around 1620 cm^−1^, C=O vibration at around 1720 cm^−1^, C-O vibration band at around 1420 cm^−1^, and epoxides at 1200–1250 cm^−1^. In addition, the vibration band at around 830 cm^−1^–850 cm^−1^ was detected especially in GO_c39_ and GO_c40_ samples. This band might be assigned as the C-Cl vibration band as reported in the other work [32]. However, the C-Cl vibration band is unexplained, because in this work, there is no other additional treatment that refers to the addition of Cl. Since the GO is a nonstoichiometric compound, the structure analysis is quite difficult, and the variety of the composition depends on the synthesis condition. It can be assumed that the remaining Cl impurity is coming from laboratory glass material. The difference in the degree of oxidation was revealed between GO_c70_, GO_c72_, and GO_c39_, and GO_c40_; i.e., an epoxide signal was seen in the GO_c39_ and GO_c40_ samples.

### 3.3. Morphology of GO

TEM imaging and SAED diffraction are used to draw the structural model in each GO sample, as shown in Figure 1. For TEM image measurement, 200 kV was applied in each sample. Under this condition, the morphologies of the GO samples were shown to be lamellar, with wrinkles and overlaps, and they were also found to be stable. The image indicates that the some GOs had been exfoliated. This is in agreement with the previous research in which both wrinkles and overlaps are essential features of GO-based bulk material [33]. The sharp spots in an SAED pointed out the crystalline region in GO. Especially GO_c72_ shows a typical diffraction pattern representing the crystalline nature of the GO sample. The present SAEDs are in agreement with the previous work that presented the amorphous region (see Figure 1(A1,C1)) and crystalline region (see Figure 1(B1)) in the GO sample [34]. Particularly, the amorphous zone is associated with the presence of the sp^2^/sp^3^ carbon cluster (see Section 3.7.2) in all the GO samples [35]. In the other imaging using SEM measurement, it is shown in Figure 1d,e that the GO sample consists of several layers (see Appendix A). Previous work emphasized that the morphology and quality of the multilayers of GO depend on several factors such as the source of graphite, degree of oxidation, and reaction condition [36]. All these GO samples are aligned with a few layers of GO in the previous work [37].

### 3.4. Morphology of GO-POM Composite and Its EDS Map

The SEM image of the GO-POM nanocomposite and its EDS map are shown in Figure 2 and Appendix A. Figure 2a shows the SEM of [GO_70_POM]_41_, and Figure 2b–d show the W, O, C element distribution, respectively. Tungsten is dominantly originated from the α-K_6_P_2_W_18_O_62_ particle, and carbon is coming from the GO. The result suggests that the tungsten, oxygen, and carbon elements were homogeneously distributed in the GO-POM nanocomposite, showing the successful combination between GO and POM [38]. This phenomenon was also confirmed by TEM measurement. Figure 2e show the homogenous POM particle in the surface area of GO after forming a composite. The SEM image of [GO_70_POM]_41_ (Figure 2e) is darker than that of the original GO_C70_ (Figure 2f). The formation of a layer after forming the composite [GO_70_POM]_41_ (Figure 2(E1)) was confirmed by comparing it with original GO_C70_ (Figure 2(F1)) in the same-scale image (200 nm).

### 3.5. Powder X-ray Diffraction Pattern (PXRD)

Powder X-ray diffraction (PXRD) measurement was carried out to characterize the GO, Dawson-type POM, and GO-POM nanocomposite structures. As shown in Section 3.7.1, Appendix A, the PXRD patterns of the GO_c70_, GO_c72_, GO_c39_, and GO_c40_ show diffractions at 2θ = 11.58°, 8.66°, 8.91°, and 9.13°, respectively. The d-spacing increases from 0.34 nm of graphite spacing (002) to 0.77–1.02 nm, which corresponds to the typical diffraction peak of the GO nanosheet (Appendix A). The interlayer distance was calculated by Bragg’s law, as shown below:d = λ/2 sin (θ) = 0.154 nm/2 sin (θ)(3)
where d is the distance between the layers of GO, θ is the diffraction angle, and λ is the wavelength of the X-ray beam (λ = 0.154 nm).

Although the d-spacing of GO_c70_ is shorter than those of GO_c72_, GO_c39_, and GO_c40_, it is suggested that oxygen-containing functional groups (epoxide, hydroxyl on the basal plane, and carboxyl groups on the edge of the basal plane) successfully formed in different degrees of oxidation. This is the reasonable explanation to the enhancement of the interlayer spacing of GO [39].

The diffraction peaks of all samples were slightly shifted by forming the composite (Appendix A). That is, the d-spacing decreased by forming the composite, except for [GO_70_POM]_41_ (Appendix A). The result of [GO_70_POM]_41_ might agree with the previous work that might be assumed as a consequence of a partial intercalation of the POM species and random arrangement of the carbon layers [40]. The results except for the [GO_70_POM]_41_ result revealed that the α-K_6_P_2_W_18_O_62_ species are not embedded in between the GO sheets; however, it could be assumed that the POM species decorated on the surface of GO, as is assumed in Appendix A. It is already known that the molecular size of the α-K_6_P_2_W_18_O_62_ species is larger (ca. 1.03 nm × 1.5 nm) [20] than the d-spacing of each GO sample (0.77–1.02 nm).

In addition, we could see in Appendix A the out of plane crystallite size of GO samples. The size was calculated by using the Debye–Scherrer equation:D (nm) = 0.9λβ^−1^ (cos (θ) )^−1^(4)
where D is the average out of plane crystallite size (nm), λ is the wavelength of the X-ray beam, β is the FWHM (full width at half maximum), θ is the XRD peak position.

The in-plane crystallite size of the GO samples and the average number of graphene oxide layers (*n*) per domain were also estimated by using PXRD data. The following general formula is described in [41]:L (nm) = 1.84 λ/β·cos(θ)(5)
*n* = D/d + 1 = (0.9λβ^−1^ (cos (θ))^−1^/d) + 1(6)
where D is the average out of plane crystallite size (nm), d is the interlayer distance, λ is the wavelength of the X-ray beam, β is the FWHM (full width at half maximum), θ is the XRD peak position, and *n* is the number of layers of the GO sample. These calculations provide crucial information regarding the average crystallite size (reflecting the enhancement or reduction of the graphitic zone and the grain boundaries or lateral defect formation) and the average number of GO layers (see Appendix A).

### 3.6. Raman Spectroscopy

Raman spectroscopy was used to characterize the GO material, and also, it provides the structural information in detail to draw its quality [42]. From the spectra, we could see the D bands located at 1356 cm^−1^, 1352 cm^−1^, 1352 cm^−1^, and 1347 cm^−1^ for GO_c70_, GO_c72_, GO_c39_, and GO_c40_, respectively (see Appendix A). This indicates the extensive oxidation of graphite and extended amounts of the sp^3^-hybridized carbon atoms [43]. The G bands located at 1586, 1581, 1590, and 1590 cm^−1^ for GO_c70_, GO_c72_, GO_c39_, and GO_c40_, respectively, correspond to the sp^2^-hybridized carbon atoms in the hexagonal framework. The G band is inherently with the 2D band (the second-order Raman modes) that appears at 2700 cm^−1^. The critical thing to fully and accurately characterize the GO sample is the consideration of the second-order Raman modes [44]. The area ratio of the 2D band also has a linear correlation with the hole mobility [45]. In addition, the previous work has explained the packing mechanism of multiple layers GO (bi-, tri- layers) by STEM-ADF image simulation. It showed that the disorder of the graphene sheet and the roughness have been created by the random covalent attachment of oxygen on the top and bottom surfaces. It influences the lattice distortion and breaks the symmetry of the system [46]. Therefore, it can be predicted that the different C/O ratio of GO samples in our work is sufficient to influence the material structure of the GO-POM composite and the ability to adsorb the Cs.

As presented in Appendix A, the intensity ratio of I_D_/I_G_ of each original material (GO_c70_, GO_c72_, GO_c39_, and GO_c40_) are 0.93, 0.91, 0.89, and 0.88, respectively. These values are correlated with the different compositions of the C/O ratio on each GO sample that are already discussed in the previous section. The intensity ratio (I_D_/I_G_) tends to increase by forming a composite. Therefore, we may conclude that the anchored process of POM on the GO has affected the structure of GO. In this case, also, there was an opposite trend for GO_c70._

### 3.7. Cs Adsorption Performance

#### 3.7.1. Adsorption Efficiency and Adsorption Capacity of GO and POM

The Cs adsorption capacities of GO_c70_, GO_c72_, GO_c39_, and GO_c40_ and POM are summarized in Table 2. Those of GO-POM nanocomposites are presented in Table 3 (see Section 3.7.2). The GO samples were classified into two types according to the elemental analysis. The GO_c70_ and GO_c72_ have 70–72 wt % C element, while GO_c39_ and GO_c40_ have ≈39 wt % C element. As shown in Table 2, the result of the present work pointed out that there is not much difference in the Cs^+^ adsorption by POM, GO_c70_, GO_c72_, GO_c39_, and GO_c40_ among the samples. The role of different C/O ratios in capturing Cs cannot be revealed. It is also already shown by other researchers that the C/O ratio of GO does not have systematic correlation with the ion exchange capacity [43].

Furthermore, in this work, the interaction between GO_c70_ and the Cs cation occurred immediately less than one minute after the stirring process (see Appendix A). A brown milky coagulation was formed. The same behavior was observed between GO_c72_ and Cs; however, there was a difference for GO_c39_: the brown milky coagulation was formed in a small quantity, and the milky coagulation was not formed for GO_c40_. The difference might be related to the variance in C/O ratio between GO_c70_, GO_c72_, and GO_c39_, GO_c40_. Moreover, it was explained that by using a surface functional group, the GO can attract the Cs cation by forming inner-sphere complexes [47]. The metal complexing in the oxidized GO strongly encourages the aggregation behavior and can be quantified by the critical coagulation concentrations (CCC) [48]. TEM measurement shows that there are many Cs clusters in the surface of GO_c70_ (see Figure 3c). Especially for this phenomenon, we also considered that the Cs cluster may include Al impurity (see Figure 3d). It might come from laboratory glass material, as reported by other work [31]. The black spots also can be seen in the [GO_40_POM]_41_, which consists of GO_c40_/POM. However, the black spots are not uniformly found all over the surface area. It is theoretically explained that most of the surface reactions eventuate only on the active site and do not happen uniformly over the surface area [49]. As proposed by other work, the edge site of graphene (carbene- and carbyne-type carbon atoms) has a significant role for adsorption on the surface of sp^2^-hybridized carbon [50]. It is also confirmed by previous study that some ions or molecules dissolved in water have a good penetration rate through the graphene nanocapillaries until the physical size of the ions or molecules exceeds the critical one [51]. A highly oxidized domain and the hydrophobic domain in GO have different roles. The highly oxidized domain would be responsible for rapid water transport in the interlayer of GO, while the hydrophobic domain would serve a network of capillaries [43]. Another researcher also considered the polarizability of graphene in order to estimate the orientation of interfacial water [52], and the interfacial water can be used to observe the surface adsorption behavior [53]. Based on these analyses, it can be assumed that the interfacial water may support the Cs interaction with GO.

The other important evidence of Cs adsorption is also visible in PXRD measurement. The sharp and intense diffraction peaks were reduced by introducing Cs (Figure 4 and Appendix A). The broad diffraction of GO_c70_, GO_c72_, GO_c39_, and GO_c40_ showed the non-uniform surface complexation with the Cs cation. PXRD and TEM show the formation of the Cs cluster. This might correspond to the origin of GO acidity, which contributes to the high cation exchange capacity (CEC). The GO acidity is made by oxygen functional groups, and the main moieties responsible for the acidic properties are carboxyl groups [43]. Moreover, according to the result by Dimiev et al. (2012), there is one acidic site for every six to eight carbon atoms; i.e., 100 g of GO sample contains 500–800 mmol of active acidic sites [54]. The adsorption capacity of the present GOs (≈3200 mmol/100 g) (Table 2) is larger than the concentration of the active acidic site. This is due to the formation of a Cs cluster. It is also generally accepted that the basicity and the size of the molecule to be adsorbed are a significant aspect in the adsorption process [55,56]. In this work, the pH of the solution is around 5–6 when the Cs contacts the GO-POM composite or GO only. Some previous researchers have proved that the pH condition in GO samples influences the adsorption capacity. It can be explained that at pH above 3.9 (pH_pzc_), the surface area of GO is partially negatively charged and the electrostatic attractions between cations and GO are stronger because of deprotonation reaction [57]. It is also caused by the higher availability of active sites on the GO sheets [58].

As discussed above, these data have a good correlation with the information that is observed in Raman spectra. Based on Raman analysis (Figure 5 and Appendix A), it was found that the change in spectrum by Cs adsorption was similar for both the GO-only samples and composite samples. It can be seen that the second-order peaks have changed after Cs adsorption. In previous studies, it was strengthened that the second-order peak indicates that the crystallite size has changed. This indicator showed that the spectrum broadens consistently with decreasing the domain size [59]. Therefore, during the chemical processing (Cs adsorption), the significant structural change of the carbon framework occurred. In addition, based on the sample experiment using [GO_39_POM]_41_, there is a slight shift in the D band from 1347 cm^−1^ before Cs adsorption to 1352 cm^−1^ after Cs adsorption. The same shift also occurred in the other samples, [GO_70_POM]_18_ and [GO_70_POM]_41_ (Appendix A). It might be caused by the interaction of oxygen functional groups and the hole defect in the GO sheets with Cs cation [58]. Thus, the change that the crystal structure after Cs adsorption has altered by amorphization could be confirmed.

#### 3.7.2. Adsorption Efficiency and Adsorption Capacity of GO-POM Nanocomposite

Table 3 shows the result of Cs adsorption by using the GO-POM nanocomposite. The GO-POM nanocomposite (Table 3) increased the Cs adsorption capacity effectively compared to the adsorption capacity of the original material (Table 2). The change range is around 6.7% to 42.9%. This result is consistent with the previous finding that signified the characteristic of surface area, showing an important role of POM species with carbon materials [60,61]. As a consequence, the Cs^+^ adsorption becomes high. We compared the average increment of Cs adsorption of the composite material between GO having 70–72 wt % C element and GO having 39–40 wt % C element. The functional groups of GO provide a binding point for chemical modification [25]. It is seen in Figure 6 that there is no significant difference between GO having 70–72 wt % C element and GO having 39–40 wt % C element.

We compared the average increment of Cs adsorption of the composite material between the concentration ratio 1:8 and concentration ratio 4:1, but we could not find the significant difference. In principle, the adsorption behavior of the GO-POM nanocomposite might be influenced by the significant role of GO as the electron acceptor and Dawson-type POM as the electron donor that led to enhance the attraction between Cs^+^ and the composite [62]. The other phenomena authenticated by another researcher confirmed that the interaction between POMs and GO can provide more mobile protons and flexible pathways [63]. In addition, Wang’s group reported that the surface charges of GO modified by functional groups (GO-COOH, GO-NH_2_, GO-OCH_3_) can be used to modulate the charge transport [64]. Therefore, the presence of Dawson-type POM in the GO system enhanced the negative charge of the GO part in the composite and provided a good adsorption of Cs^+^ ion (Appendix A). The interaction of Cs^+^ ion with Dawson-type POM was an ion-pair complex [65]. The anion surface of Dawson-type POM is electron rich and has a strong interaction with Cs^+^. Therefore, the introduction of Dawson-type POM in our adsorbent system was important to enhance the adsorption of the Cs^+^ ion. However, further investigation is still needed to find out the mechanism that occurred behind the phenomenon.

It is also reported that the mechanism of cation–GO interaction in the adsorption behavior should be considered together with aggregation process [66]. In the present study, Figure 7a,b exhibited that the aggregation occurred after Cs adsorption by using the GO-POM nanocomposite. The results demonstrated that the composite samples immediately coagulate after Cs adsorption except for [GO_70_POM]_41_, which still forms a stable solution after one hour keeping at ambient temperature (see Appendix A). We predicted that a small granule of Cs was formed, and there was a crinkled GO sheets coating around it (see Appendix A). It can be signified that a stronger bond was formed. It might be predicted that there is a different composition of oxidative debris (OD) component among the GO samples. As a previous study explained, the OD of GO has a significant role in the origin of GO acidity [67]. This is already discussed in the previous section (Section 3.7.1 regarding the contribution of GO acidity to the cation exchange capacity). However, further investigation is still needed to prove it and to obtain strong evidence. Unfortunately, the elemental analysis of Appendix A was not conducted. Regarding this phenomenon, it can be assumed that there are two important factors that might contribute to the aggregation behavior. First, the size of the GO and composite sample matter. As presented in Appendix A, the size of each sample is diverse. The range of values is around 4.3 to 34.78 nm. A previous study showed that the size of GO (nano GO, colloid GO, micron GO) can strongly influence the aggregation behavior [68]. Second, the other factor is the degree of acidity. It is strongly related with the functionalities of the GO sample. It was explained in an earlier section in this work that there are two types of GO samples, which have different C/O compositions. Evidence was shown by the previous study that the GO (single or multilayers) coagulates in highly acidic conditions because of losing their surface charge [54]. It is also correlated well with other work that identified the effect of cations on aggregation [47]. Moreover, it was strengthened in the previous theoretical analysis that considered the important role of H_2_O as a dipole molecule involving cation–π interaction. Therefore, the electrostatic interaction would be significantly influenced [69,70]. As calculated in the previous studies, the value of the internuclear distances d_ion-water_ of Cs^+^ is 0.315 nm [71]. Therefore, the interaction of Cs^+^ with the GO-POM nanocomposite becomes easier to diffuse through the interlayer distance of the GO-POM nanocomposite, which has a d-spacing around 0.69–0.81 nm and 0.77–1.02 nm for the GO sample (see Appendix A).

By FTIR spectra, specific changes are marked at 1150–1350 cm^−1^ after Cs adsorption in all GO-POM nanocomposites. Especially for composites that consist of ≈39 wt % C element, the IR spectrum shows more significant changes (see Figure 8e–h and Appendix A), whereas for the composites that consist of 70–72 wt % C element, the peak shifts are not so significant, thus making analysis quite difficult (see Figure 8a–d and Appendix A). A similar tendency also occurred in all GO samples after Cs adsorption (see Appendix A). This result indicates that there is a change in the interlayer chemistry of the sample that influences the defect (hole) formation and functional group formation. It might be caused by the interaction between the GO functional group (including carboxyl, phenol, multi-group ether) and the Cs cation [58]. As explained by another researcher, the pH-driven reversible epoxide formation in GO (epoxy opening/closing reactions) should be considered as an important part of the reactivity and properties of GO [72]. The tendency is observed that the interlayer distance after Cs adsorption in all nanocomposite samples were expanded, even though the increment was not so high (see Appendix A). For example, in the [GO_40_POM]_18_, [GO_40_POM]_41_, [GO_39_POM]_18_, and [GO_39_POM]_41_ composites, the interlayer distances were changed from 7.6 to 8.3 A, 7.6 to 7.8 A, 7.4 to 8.1 A, and 7.4 to 7.9 A, respectively. It also happened in the [GO_70_POM]_18_, [GO_70_POM]_41_, [GO_72_POM]_41_, and [GO_39_POM]_18_ composites, where the d-spacing was expanded from 7.2 to 7.9 A, 8.1 to 8.6 A, 7.3 to 7.9 A, and 6.9 to 7.7 A, respectively. In addition, the other study confirmed that the CO_2_ gas is trapped within the multilayers of GO (marked at 2357–2359 cm^−1^). Intentionally, they verified that the intercalated water has an essential role for accelerating the reaction by some specific mechanism [73].

Another work also confirmed that the functional group of GO can act as a ligand to replace the water molecule from the metal cations in the first coordination sphere. Previous work verified that the relaxation time by using the NMR relaxation method can be considered as an essential variable for identifying the metal complexing mechanism of GO in different charge, size, and electron configuration of the metal as a function of pH and GO concentration [74]. It was also confirmed by another study that the presence of wrinkle-like water tunnels has a strong relation with the water adsorption ability of GO [75].

Deeper investigation into carbon species in each composite sample was performed. This is shown in Table 4, where the graphitic zone can be calculated by using the integrated intensity of graphitic (2θ = 26.6) and oxidized peaks (2θ = 8.66–15) in the PXRD pattern. The amount of graphitic zone (*G*%, unoxidized region) was calculated using Equation (7), as follows:(7)$$G\%=\;\frac{Igraphite}{Igraphite+Ioxidized}\;\times \;100$$
where *I_graphite_* and *I_oxidized_* are the intensities of the graphitic and oxidized diffraction peaks of GO in the PXRD pattern, respectively [76].

As presented in Table 4, the diversification of sp^2^/sp^3^ carbon clusters on each sample has been produced. This variance among the samples indicates that there is a difference in the degree of oxidation. However, a peculiar phenomenon was seen in the GO_c70_ sample. The data in Table 4 show that the percentage of the oxidized zone of GO_c70_ is higher compared to those of GO_c72_, GO_c39_, and GO_c40_. The ratio of the oxidized zone is not much different for GO_c72_, GO_c39_, and GO_c40_. The GO_c70_ shows the shorter interlayer distance than GO_c72_, GO_c39_, and GO_c40_. This may cause the peculiarity of GO_c70_. However, the peculiarity of GO_c70_ is still under study. According to the literature, the ratio of the oxidized zone does not have a linear correlation with the oxidation ratio [77]. The unusual characteristic of GO_c70_ also has a direct effect on the composite material produced by using GO_c70_ (as provided at the beginning of Section 3.7.2). Further investigation is needed to obtain more understanding of the dependence of these factors. A recent study revealed that the ratio C/O has a big influence on the physical properties because of the change of the surface speciation [78]. Another investigation also confirmed a strong correlation between the content of Lewis acid sites (via basal plane epoxide) and the Bronsted acidic carboxylic group in the GO sample with a strong acidity and high oxidation degree [79].

Furthermore, the data show that the oxidized zone (sp^2^/sp^3^ hybrid carbon) of each composite sample was enlarged compared to the corresponding original GO. Only [GO_70_POM]_41_, which consists of GO_c70/_POM with a concentration ratio of 4:1, was decreased. It correlated well with the in-plane crystallite size of each composite material that expanded (see Appendix A). Particularly, the GO-POM nanocomposite with a concentration ratio of 4:1 shows an in-plane crystallite size larger than the GO-POM nanocomposite with a concentration ratio of 1:8. It could be assumed that there is an electron transfer from POM species to GO [80,81]. Another researcher also confirmed that the surface area of the GO-POM composite was improved five times due to the covalent bond between the POM species and GO [38]. In general, the enhancement of adsorption capacity is in line with the increase of sp^2^/sp^3^ carbon clusters on each composite surface. Based on the result in Figure 6, the lowest value of the percentage increase of Cs adsorption capacity is observed in [GO_70_POM]_41_. It is only 6.7%. This might be caused by the decreased oxidized zone (sp^2^/sp^3^ hybrid carbon) after the composite was formed (see Table 4). Table 4 shows that the value decreased from 30.3% (GO_c70_) to 28.6% (GO_c70_/POM composite). It is also confirmed by the PXRD pattern, which shows the same tendency in [GO_70_POM]_41_: that the diffraction has reduced from 11.57° to 10.99° if compared with GO_c70_ as a precursor material. Some researchers have tried to address the complexity of GO especially to produce a graphene-based hybrid material for diverse applications. For example, one researcher has attempted to control the sp^2^/sp^3^ hybrid carbon structure in GO by a simple ethanol solvothermal method [82]. Furthermore, a similar trend was investigated to enlarge the carbon cluster size by increasing the sonication time [83].

## 4. Conclusions

In this work, the GO-based POM nanocomposite was successfully obtained. The Cs adsorption capacity increased by forming a nanocomposite. The result proposed, first, that the C/O ratio of the GO sample has significantly influenced the characteristic of the GO-POM nanocomposite for the Cs adsorption performance after forming the composite. Specifically, the GO sample, which consists of 70–72 wt % C element, is a good precursor material to incorporate Dawson-type POM compared to the GO that has ≈39 wt % C element in their atomic composition. Second, the concentration ratio of GO/POM in forming a composite material can be considered to maximize the adsorption performance. Additionally, the size and the acidity of the GO also influence the aggregation state. However, further investigation is needed to know the size tunability and the role of acidic properties of GO in different C/O compositions for Cs adsorption. The further calculation was performed to quantify the graphitic zone in each sample. The result revealed that the oxidized zone was enhanced after forming a composite in all samples except [GO_70_POM]_41_, which shows a different character. Further scrutiny is needed to open the understanding of the role of the oxidative debris (OD) component of GO and the level of hydration of GO for adsorption capacity.

## Figures and Tables

**Figure 1 materials-14-05577-f001:**
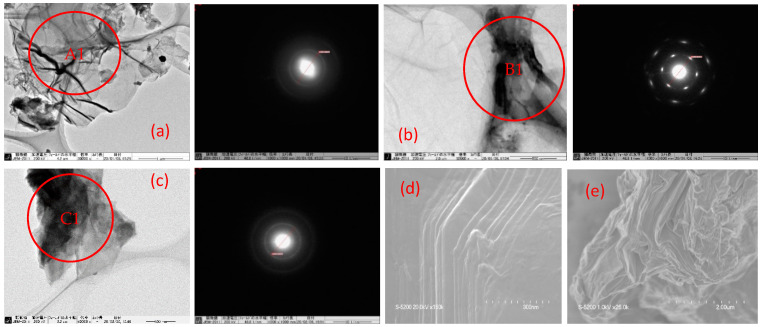
TEM imaging of (**a**) GO_c70_, (**b**) GO_c72_, and (**c**) GO_c39_ and SEM imaging of (**d**) GO_c39_, (**e**) GO_c40_, and SAED pattern from the location (**A1**,**B1**,**C1**) marked in pictures (**a**–**c**), respectively.

**Figure 2 materials-14-05577-f002:**
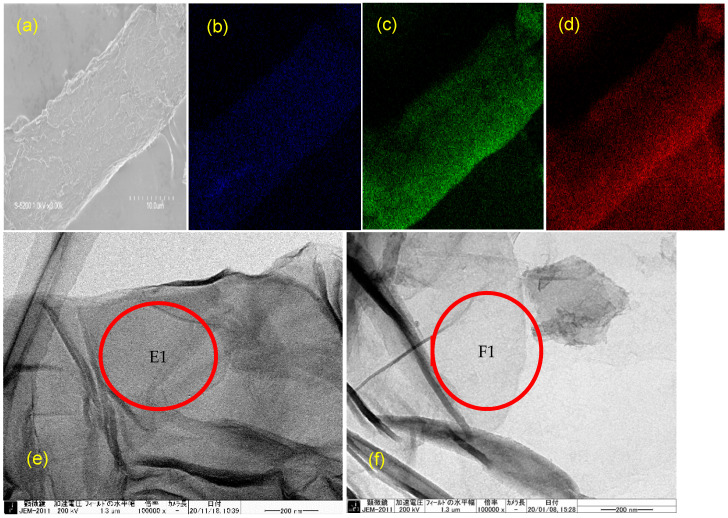
SEM measurement: (**a**) [GO_70_POM]_41_, its EDS map (**b**) Tungsten element, (**c**) Oxygen element, (**d**) Carbon K-α X-ray element; TEM image: (**e**) [GO_70_POM]_41_, (**f**) GOc_70_, showing the formation of (**a**) layer after forming the composite [GO_70_POM]_41_ (**E1**) compared with the original GO_C70_ (**F1**) in the same-scale image (200 nm).

**Figure 3 materials-14-05577-f003:**
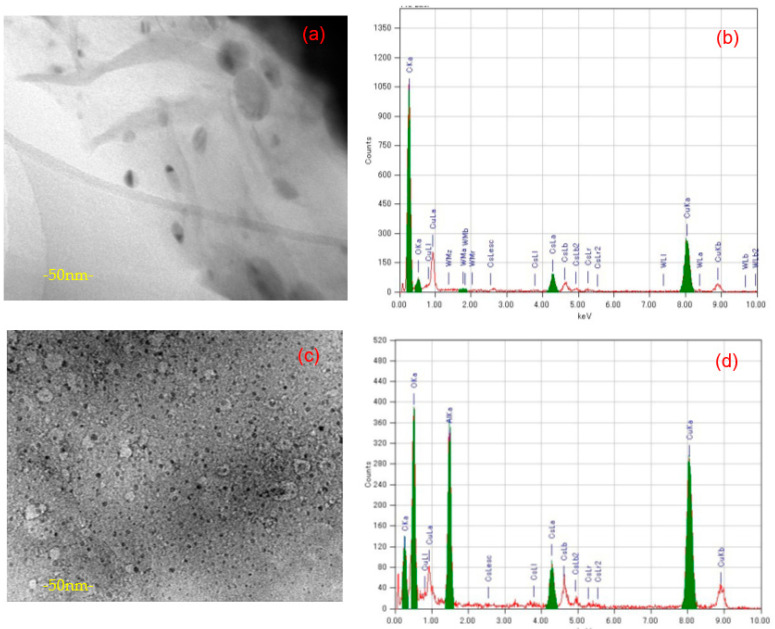
TEM image: (**a**) [GO_40_POM]_41_ after Cs adsorption, small black spots are identified (scale: 50 nm), (**b**) Elemental analysis of (**a**), (**c**) GO_c70_ after Cs adsorption, many small Cs clusters are seen (scale: 50 nm), (**d**) Elemental analysis of (**c**).

**Figure 4 materials-14-05577-f004:**
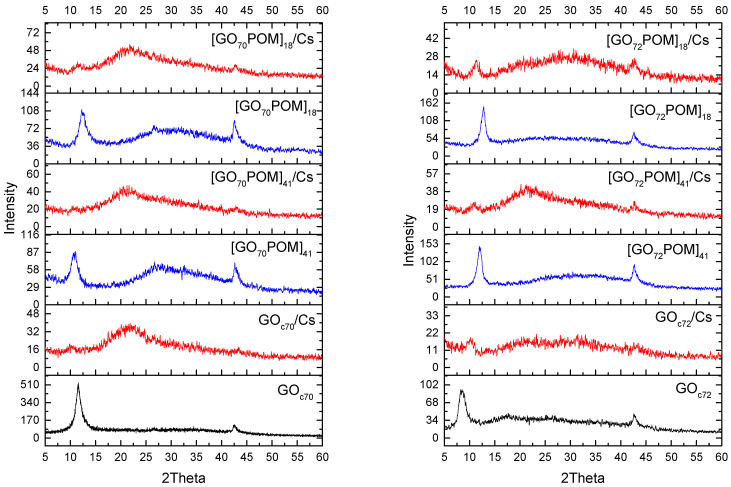
PXRD patterns of GO_c70_, GO_c72_, and their GO-POM nanocomposite before and after Cs adsorption.

**Figure 5 materials-14-05577-f005:**
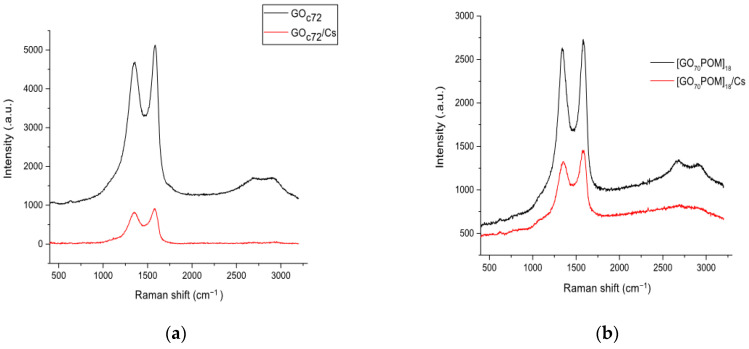
Raman spectra before and after Cs adsorption for (**a**) GO_c72_ (**b**) [GO_70_POM]_18_.

**Figure 6 materials-14-05577-f006:**
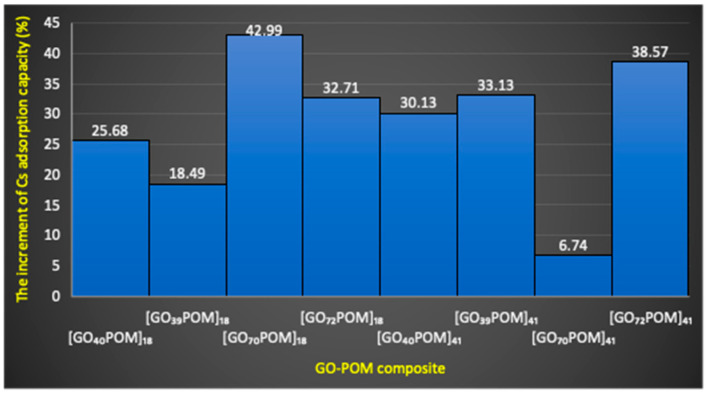
The increment of Cs adsorption capacity (%) calculated by using the ratio before and after forming a composite in each GO sample.

**Figure 7 materials-14-05577-f007:**
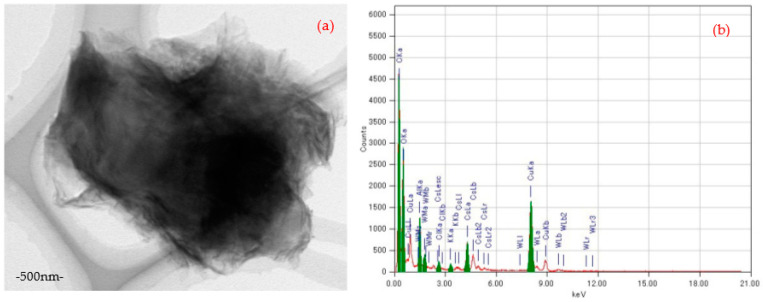
TEM image of [GO_70_POM]_18_/Cs: (**a**) Aggregation after Cs adsorption, (**b**) Elemental analysis of (**a**).

**Figure 8 materials-14-05577-f008:**
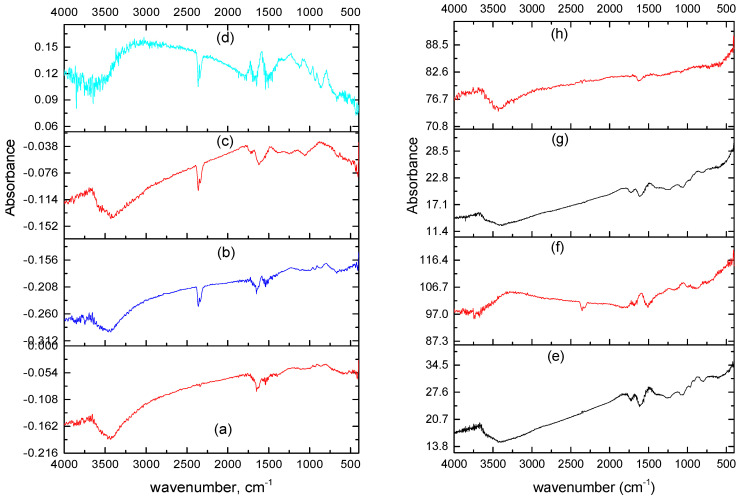
IR spectra for (**a**) [GO_70_POM]_18_, (**b**) [GO_70_POM]_18_ after Cs adsorption, (**c**) [GO_70_POM]_41_, (**d**) [GO_70_POM]_41_ after Cs adsorption, (**e**) [GO_40_POM]_18_, (**f**) [GO_40_POM]_18_ after Cs adsorption, (**g**) [GO_40_POM]_41_, and (**h**) [GO_40_POM]_18_ after Cs adsorption.

**Table 1 materials-14-05577-t001:** Elemental analysis of GO sample.

Sample	C%	H%	N%	O%	C/O
GO_c70_	70.87	7.95	-	21.18	3.34
GO_c72_	72.04	8.07	-	19.89	3.61
GO_c39_	39.28	2.76	0.27	57.69	0.68
GO_c40_	39.61	2.58	0.25	57.56	0.69

**Table 2 materials-14-05577-t002:** Adsorption capacity, adsorption efficiency, and adsorption condition.

Materials	Adsorption Efficiency	Adsorption Capacity	Stirring	Sonification Time
	(%)	(mmol/g)	Time	(minute)
POM-Cs	58.3	32.8	24 h	5
GO_c70_-Cs	58.8	33.1	24 h	5
GO_c72_-Cs	56.8	31.9	24 h	5
GO_c39_-Cs	61.3	34.5	24 h	5
GO_c40_-Cs	56.3	31.7	24 h	5

**Table 3 materials-14-05577-t003:** Adsorption capacity, adsorption efficiency, and adsorption condition.

Materials	Composition	Concentration Ratio	Adsorption	Adsorption Capacity	Stirring	Sonification Time
		GO:POM	Efficiency (%)	(mmol/g)	Time	(minute)
[GO_70_POM]_18_	(GO_c70_/POM)	1:08	84	47.3	24 h	5
[GO_70_POM]_41_	(GO_c70_/POM)	4:01	62.7	35.3	24 h	5
[GO_72_POM]_18_	(GO_c72_/POM)	1:08	75.4	42.4	24 h	5
[GO_72_POM]_41_	(GO_c72_/POM)	4:01	78.7	44.3	24 h	5
[GO_39_POM]_18_	(GO_c39_/POM)	1:08	72.7	40.9	24 h	5
[GO_39_POM]_41_	(GO_c39_/POM)	4:01	81.7	45.9	24 h	5
[GO_40_POM]_18_	(GO_c40_/POM)	1:08	70.7	39.8	24 h	5
[GO_40_POM]_41_	(GO_c40_/POM)	4:01	73.5	41.3	24 h	5

**Table 4 materials-14-05577-t004:** The sp^2^/sp^3^ carbon cluster on each composite surface.

Material	Composition	Graphitic Zone (G%)	Oxidized Zone (%)
GO_c70_	-	69.7	30.3
GO_c72_	-	75.4	24.6
GO_c39_	-	74.9	25.1
GO_c40_	-	74.5	25.5
[GO_70_POM]_18_	GO_c70_/POM, 1:8	68.4	31.6
[GO_70_POM]_41_	GO_c70_/POM, 4:1	71.4	28.6
[GO_72_POM]_18_	GO_c72_/POM, 1:8	67.5	32.5
[GO_72_POM]_41_	GO_c72_/POM, 4:1	69.0	30.9
[GO_39_POM]_18_	GO_c39_/POM, 1:8	69.0	31.0
[GO_39_POM]_41_	GO_c39_/POM, 4:1	69.3	30.7
[GO_40_POM]_18_	GO_c40_/POM, 1:8	69.1	30.0
[GO_40_POM]_41_	GO_c40_/POM, 4:1	69.5	30.5

## Data Availability

Data sharing is not applicable for this article.

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
