# Peer review of "Exploration of the Cs Trapping Phenomenon by Combining Graphene Oxide with α-K6P2W18O62 as Nanocomposite"

_materials, 2021, doi:10.3390/ma14195577_

Round 1

Reviewer 1 Report

See attached file.

Author Response

Response to Reviewer 1

We thank the reviewer for a lot of the constructive and fruitful comments for our manuscript. We have revised our manuscript by referring to the reviewer’s comments as long as we can. The revisions are performed as follows:

No.

Reviewer/editor

Input

Response

1.

Reviewer 1

1. The scale information of each SEM/TEM image is not clear and needs to be edited. The resolution of Fig. 3b and Fig. 3d is also too low that the labels on the peak can be hardly seen.

Thank you for the comment. We changed the order of 3a, 3b and 3c, 3d. We changed the file from (.emsa) to be (.postscript) to keep the resolution.

2. The discussion sections need to be better written in general in terms of avoiding redundant information. For example, on line 341, “This shows the formation of Cs cluster” should also be removed as it is essentially a repetition of its previous sentence.

Thank you for the comment. We checked the whole manuscript and then we avoided the redundant information as much as we can.

3. For the discussion in Section 3.3, first of all, Fig. 1c is missing. Second, the authors stated “This is in agreement with the previous work that presented the amorphous region and crystalline region in GO sample.” However, Fig. 1 only shows the crystalline region, while no amorphous region is marked. I suggest the authors add the amorphous region characterization here. Finally, by comparing these GO samples with different C/O ratio, is there any structural difference that the authors can further address?

Thank you for the comment. There is Fig. 1c below Fig. 1a. In addition, the amorphous region of GO sample was signed in Fig. 1 (A1), (C1). We added corresponding section in the discussion below Fig. 1 as below. “The present SAEDs are in agreement with the previous work that presented the amorphous region (see Figure 1(A1),(C1)) and crystalline region (see Figure 1(B1) in GO sample [30].”Regarding the structural difference among the GO samples, it`s still need further investigation to identify each GO sample. We tried to add additional explanation in the section 3.1 as below. “In particular, it can be assumed that there is a difference in structure between the two types of GO. The GOc70 and GOc72 indicating few oxygen defects in the structure have an important contrast with the GOc39 and GOc40.However, deeper characterization is still required since the oxygen defect is not only the case. As emphasized in the previous literature, there is no evident relation between oxidation degree and ordered domain that is exhibited in Raman spectrum [13].”  

4. In section 3.4, what is the purpose of showing Fig. 2e here? It is not mentioned in the main text and didn’t provide more information to the analysis. I suggest to either remove Fig. 2e or put it in the supporting information.

In the same section, the authors mentioned “Figure 2 (f) and (g) show the homogenous POM particle in the surface area of GO after forming composite. “ For me it’s hard to tell the POM particles from these two pictures. I suggest the authors either mark the particles in the figure, or provide a TEM image at the same scale of GO samples to show the difference.  

Thank you for the comment. We have checked and revised it based on the referee’s suggestion. We removed Fig. 2e and we added one image (GOc70) as new Fig. 2e, which is the image of TEM measurement with similar scale compared to the composite image in order to see an appearance of the layer between both images (before and after forming the composite)

5. In section 3.6, the authors listed peaks for each GO samples where peak shifts are observed. So what does these shifts in Raman spectra mean? What's the effect of C/O ratio to the spectra/materials structure? I hope the authors can provide a deeper discussion on the difference of these samples.

Thank you for the comment. Based on that, we have added some explanation regarding the effect of C/O ratio to the material structure. The previous works explained the packing mechanism of multiple layers for GO by STEM-ADF image simulation. We added as below.

“It showed that the disorder of graphene sheet and the roughness have been created by the random covalent attachment of oxygen on the top and bottom of surfaces. It influences the lattice distortion and breaks the symmetry of the system [80]. Therefore, it can be predicted that the different C/O ratio of GO samples in our work is sufficient to influence the material structure of the GO/POM composite and the ability to adsorb the Cs.”

6. In section 3.7.1, line 319, the author mentioned “their GOPOM nanocomposites were analysed.” This analysis actually comes in section 3.7.2, so I suggest the authors to add the corresponding section/table information after this sentence to better guide the readers.

In the same section, I don’t think it’s necessary to list the adsorption capacity of each GO sample in the text, since the readers can clearly see the numbers from Table 1. And the sentence of “The adsorption capacity can be ranked as follows GOc39 > GOc70 > POM > GOc72 > GOc40.” should be removed, because the main conclusion the authors want to draw here is that “there is not much difference among 326 the samples.” 

Thank you for the comment. We added corresponding section as suggested.

“The Cs adsorption capacity of GOc70, GOc72, GOc39, and GOc40 and POM are summarized in Table 2. Those of GO-POM nanocomposites are presented in Table 3 (see section 3.7.2).”

Also, we decided to remove some unnecessary sentences as the referee suggested. We removed ““The adsorption capacity can be ranked as follows GOc39 > GOc70 > POM > GOc72 > GOc40.”

7. In section 3.7.2, the author concludes “Fig. 6 presents that GO-POM ratio 4:1 ratio tends to show higher increment than 1:8 ratio, except for [GO70POM]18 and [GO70POM]41.” This doesn’t seem to be a correct conclusion of Fig. 6, as the average increment of the four GO-POM samples with ratio 1:8 is higher than that of samples with ratio 4:1. Also, Fig. 6 needs to have a y-axis label.

Thank you for the comment. We deleted complex discussion. We just mentioned that the GO-POM nanocomposite (Table 3) increased the Cs adsorption capacity effectively compared to the adsorption capacity of original material (Table 2).

Also, we added y-axis label in fig.6 as referee suggested.

Reviewer 2 Report

The paper is aimed at development of a novel sorbent graphene oxide-based nanomaterial which is able to capture cesium with high efficiency. The subject is within the scope of MDPI Materials.

The authors have successfully synthesized and thoroughly examined the properties of eight compounds based on graphene oxide, POMs in the context of Cs sorption performance. The morphology of the examined compounds was examined using TEM and SAED diffraction. Powder X-ray diffraction measurements and Raman spectroscopy were used to characterize the examined materials.  The paper is self-contained and may be published “as it is”. This paper may attract the attention of materials scientists working with radio nuclides. It has a strong applicative character.

The authors may review their manuscript as far as the language is concerned. In particular some stylistic improvements might be applied, for example the sentences should not start with “And”.

Author Response

Response to Reviewer 2

We thank the reviewer for a lot of the constructive and fruitful comments for our manuscript. We have revised our manuscript by referring to the reviewer’s comments as long as we can. The revisions are performed as follows:

No.

Reviewer/Editor

Input

Response

1

Reviewer 2

1.The authors may review their manuscript as far as the language is concerned. In particular some stylistic improvements might be applied, for example the sentences should not start with “And”

Thank you for the comment. We have checked our manuscript again carefully.

2.The quality of some Figures is rather poor, for example Fig. 6, which is a screenshot from a spreadsheet. If possible, better resolution should be used for the photos, e.g. Fig. 7. The corresponding dependence of count vs. energy is hardly legible (Fig. 7b).

Thank you for the comment. We changed the figure 7b by changing the format from .emsa to be .postscript in order to keep the resolution. In addition, as suggested, we also have changed the Fig.6.

The authors use several words to name basically the same process “trapping/adsorption/sorption” – this should be unified.

Thank you for the comment. We changed “sorption” to “adsorption”. We left “trapping” because we used it in the title and we put the meaning of adsorption and aggregation into “trapping”.

Reviewer 3 Report

How to handle radioactive cesium is an important research topic for environment remediation. In view of the rational nanocomposite design and Cs adsorption performance, this paper is recommended for publication after addressing some minor issues.

  1. The interaction between GO and POM should be explained in detail.
  2. Apart from negative charge, mechanism of high Cs adsorption capability is still unclear for GO/POM composite. Why did the introduction of POM effectively enhance the Cs adsorption?
  3. Peaks at 2357 cm-1- 2359 cm-1 in FTIR was thought to be related to Cs adsorption. According to previous work, these peaks were ascribed to CO2. Please check this issue.
  4. Written this sentence “GO sample, which consists of a high C element”. “a high C element” is not a professional description.
  5. The stability pH values of Dawson type POM should be clarified. As is known, POM might undergo decomposition in water.

Author Response

Response to Reviewer 3

We thank the reviewer for a lot of the constructive and fruitful comments for our manuscript. We have revised our manuscript by referring to the reviewer’s comments as long as we can. The revisions are performed as follows:

No.

Reviewer/Editor

Input

Response

1

Reviewer 3

1. The interaction between GO and POM should be explained in detail.

Thank you for the comment. We tried to put additional explanation in the discussion as below. “The interaction of Cs+ ion with Dawson-type POM was an ion-pair complex [82]. The anion surface of Dawson-type POM is electron rich and has a strong interaction with Cs+. Therefore, the introduction of Dawson-type POM in our adsorbent system was important to enhance the adsorption of Cs+ ion.”

2. Apart from negative charge, mechanism of high Cs adsorption capability is still unclear for GO/POM composite. Why did the introduction of POM effectively enhance the Cs adsorption?

Thank you for the comment. We really considered about it. We have already added additional explanation based on other reference, “The interaction of Cs+ion with Dawson-type POM was an ion-pair complex [82]. The anion surface of Dawson-type POM is electron rich and has a strong interaction with Cs+. Therefore, the introduction of Dawson-type POM in our adsorbent system was important to enhance the adsorption of Cs+ion.”   

3. Peaks at 2357 cm-1- 2359 cm-1 in FTIR was thought to be related to Cs adsorption. According to previous work, these peaks were ascribed to CO2. Please check this issue.

Thank you for the comment. We have checked the previous works carefully. We added as below. “In addition, the other study confirmed that the CO2 gas is trapped within the multilayers of GO (marked at 2357 cm-1 – 2359 cm-1). Intentionally, they verified that the intercalated water has an essential role for accelerating the reaction by some specific mechanism [83].”

4. Written this sentence “GO sample, which consists of a high C element”. “a high C element” is not a professional description.

Thank you for the comment. We have changed the term by using GO which has 70-72 wt% C element and ~39 wt% C element.

5. The stability pH values of Dawson type POM should be clarified. As is known, POM might undergo decomposition in water.

Thank you for the comment. We have tried to discuss it more detail in the revised-manuscript as below. Contant’s group reported the hydrolytic stability of Dawson-type POM depending on the acidity of the solution. That is, Dawson-type POM was stable at pH lower than 6 and when the pH increased above 6, the formation of lacunary species occurs [81]. In the case of our system, preparation and adsorption experiments were performed at pH lower than 5. Therefore, the Dawson-type POM was stable in the present solution.

Round 2

Reviewer 1 Report

I'm satisfied with the changes made by the authors.